# Serum PD-1/PD-L1 Levels, Tumor Expression and PD-L1 Somatic Mutations in HER2-Positive and Triple Negative Normal-Like Feline Mammary Carcinoma Subtypes

**DOI:** 10.3390/cancers12061386

**Published:** 2020-05-28

**Authors:** Catarina Nascimento, Ana Catarina Urbano, Andreia Gameiro, João Ferreira, Jorge Correia, Fernando Ferreira

**Affiliations:** 1CIISA—Centro de Investigação Interdisciplinar em Sanidade Animal, Faculdade de Medicina Veterinária, Universidade de Lisboa, Avenida da Universidade Técnica, Lisboa 1300-477, Portugal; catnasc@fmv.ulisboa.pt (C.N.); curbano.ms@gmail.com (A.C.U.); agameiro@fmv.ulisboa.pt (A.G.); jcorreia@fmv.ulisboa.pt (J.C.); 2Instituto de Medicina Molecular, Faculdade de Medicina, Universidade de Lisboa, Lisboa 1649-028, Portugal; hjoao@medicina.ulisboa.pt

**Keywords:** feline mammary carcinoma, PD-1, PD-L1, CTLA-4, TNF-α, biomarkers, immunotherapy

## Abstract

Tumor microenvironment has gained great relevance due to its ability to regulate distinct checkpoints mediators, orchestrating tumor progression. Serum programmed cell death protein-1 (PD-1) and programmed death ligand-1 (PD-L1) levels were compared with healthy controls and with serum cytotoxic T-lymphocyte-associated antigen 4 (CTLA-4) and tumor necrosis factor-alpha (TNF-α) levels in order to understand the role of PD-1/PD-L1 axis in cats with mammary carcinoma. PD-1 and PD-L1 expression was evaluated in tumor-infiltrating lymphocytes (TILs) and cancer cells, as the presence of somatic mutations. Results showed that serum PD-1 and PD-L1 levels were significantly higher in cats with HER2-positive (*p* = 0.017; *p* = 0.032) and triple negative (TN) normal-like mammary carcinomas (*p* = 0.004; *p* = 0.015), showing a strong positive correlation between serum CTLA-4 and TNF-α levels. In tumors, PD-L1 expression in cancer cells was significantly higher in HER2-positive samples than in TN normal-like tumors (*p* = 0.010), as the percentage of PD-L1-positive TILs (*p* = 0.037). *PD-L1* gene sequencing identified two heterozygous mutations in exon 4 (A245T; V252M) and one in exon 5 (T267S). In summary, results support the use of spontaneous feline mammary carcinoma as a model for human breast cancer and suggest that the development of monoclonal antibodies may be a therapeutic strategy.

## 1. Introduction

Breast cancer is the most diagnosed tumor among women [1] and feline mammary carcinoma (FMC) is the third most common tumor in the cat. At the molecular level, there are distinct subtypes of breast cancer: Luminal A, Luminal B, HER2-positive and triple-negative (normal-like and basal-like) [2]. The HER2-positive subtype is characterized by a HER2 overexpression and a lack of hormone receptors (estrogen receptor (ER) and/or progesterone receptor (PR)), while the triple negative tumors are defined by the absence of ER, PR and HER2 expression, each representing 15–20% of all breast cancer cases [1,3]. As in human breast cancer [4], FMC presents an aggressive and infiltrative behavior [5,6], with both HER2-positive and triple negative (TN) subtypes showing worse prognosis than luminal A and B subtypes. Furthermore, a surgical approach is often necessary for the treatment of FMC, as the adjuvant chemotherapy (cyclophosphamide, vincristine, doxorubicin) is not useful in some cases [7]. Thus, the identification of novel diagnostic biomarkers and therapeutic targets is needed, not only to improve the clinical outcome of cats with mammary carcinoma but also because FMC shares clinicopathological, histopathological and epidemiological features, as well as the molecular classification with human breast cancer [4,8,9,10]. Moreover, the use of animal models has been increasing, allowing researchers to understand the mechanisms underlying tumorigenesis; mouse species are the most used due to their small size and short gestation period. Nevertheless, laboratory rodents have several limitations, such as a lack of genetic heterogeneity and discrepancies to the human tumor development [11], while pets that spontaneously develop malignant tumors provide natural models, making them excellent for the study of human breast cancer [12].

In humans, the role of the PD-1/PD-L1 axis is being investigated in different tumor types, with the programmed death ligand-1 (PD-L1) expressed by tumor cells regulating T cell activity, promoting immune suppression and tumor escape [13,14,15]. This immune checkpoint molecule can be expressed by cancer cells and immune cells, as T and B lymphocytes, macrophages and dendritic cells, being its expression on tumor-infiltrating lymphocytes (TILs) correlated with clinical response to anti-PD-L1-targeted immunotherapy (atezolizumab) [16]. Accordingly, a monoclonal antibody that blocks PD-L1 binding to the programmed cell death protein-1 (PD-1) receptor on T cells was recently approved by the Food and Drug Administration (FDA) to treat different tumor types and also PD-L1 positive unresectable locally advanced and metastatic TN breast cancer [17,18], since, several studies showed that PD-1 and PD-L1 are overexpressed in the most aggressive cancer subtypes (HER2-positive and triple negative) [19,20,21,22].

The less studied serum PD-1 (sPD-1) and PD-L1 (sPD-L1) levels may also play an important role in anti-tumor immune responses. Studies in patients with breast cancer, lung cancer, metastatic melanoma and renal cell carcinoma reported higher sPD-1 or sPD-L1 levels than in healthy controls, suggesting that elevated serum PD-1 and PD-L1 levels may promote immunosuppression and are therefore to be considered adverse prognostic factors [23,24,25]. Furthermore, cytotoxic T-lymphocyte-associated antigen 4 (CTLA-4) is another immune mediator that inhibits T-cell immune function, being also targeted in breast cancer. Accordingly, serum CTLA-4 levels are increased in cats with mammary carcinoma, as well as in human malignant tumors [10]. Some trials are evaluating the combination of anti-PD-1/PD-L1 antibodies with CTLA-4 inhibitors [17,26]. In humans, CTLA-4 regulates T-cell proliferation at the initial stage of immune responses, primarily in lymph nodes, whereas PD-1 suppresses T cells in later stages, typically in peripheral tissues. Additionally, it has been described that certain pro-inflammatory cytokines, such as tumor necrosis factor-alpha (TNF-α), can up-regulate PD-L1 expression in cancer cells, contributing to create an immunosuppressive tumor microenvironment [27,28]. Furthermore, TNF-α is described to be elevated in cats with mammary carcinoma, being positively correlated with serum CTLA-4 levels, as in humans [10].

The *PD-L1* gene encodes for a transmembrane glycoprotein with 290 amino acid residues [29], presenting two extracellular immunoglobulin domains, a cytoplasmic domain and shows a sequence homology of 74.5% with feline *PD-L1* gene (UniProt, accession numbers: *Felis catus*—M3WAP8, *Homo sapiens*—Q9NZQ7).

Considering the above evidences and the lack of knowledge on the role of the PD-1/PD-L1 signaling pathway in tumor progression in the cat, this study aimed to: (i) quantify and compare the sPD-1 and sPD-L1 levels in animals with different mammary carcinoma subtypes and healthy controls; (ii) evaluate the PD-1 and PD-L1 expression in tumor-infiltrating lymphocytes (TILs) and cancer cells; (iii) test for statistical associations between serum PD-1, PD-L1, CTLA-4 and TNF-α levels; (iv) identify genomic mutations in the *PD-L1* gene to validate future checkpoint-blocking therapies.

## 2. Results

### 2.1. Cats with HER2-Positive or TN Normal-Like Mammary Carcinoma Showed Higher Serum PD-1 and PD-L1 Levels

Serum PD-1 and PD-L1 levels were measured in cats with mammary carcinoma, grouped according to their tumor subtype and compared with serum levels detected in the healthy control group (Table 1 and Table 2).

Results from the Kruskal–Wallis test found significant differences between the mean ranks of at least one pair of groups (*p* = 0.044 for PD-1; *p* = 0.006 for PD-L1). Indeed, cats showing HER2-positive or TN normal-like mammary carcinoma displayed higher serum PD-1 levels than healthy group (1148.9 pg/mL vs. 534.0 pg/mL, *p* = 0.017; 3655.1 pg/mL vs. 534.0 pg/mL, *p* = 0.004, Figure 1A), as well as serum PD-L1 levels (5490.4 pg/mL vs. 1835.5 pg/mL, *p* = 0.032; 3641.4 pg/mL vs. 1835.5 pg/mL, *p* = 0.015, Figure 1B). Furthermore, the optimal cut-off value of serum PD-1 levels was 801.6 pg/mL in cats with HER2-positive mammary carcinoma (AUC = 0.765 ± 0.104, 95% CI: 0.562–0.968, *p* = 0.031; sensitivity = 54.5%; specificity = 91.7%; Figure 1C) and 801.6 pg/mL in cats with TN normal-like mammary carcinoma (AUC = 0.857 ± 0.092, 95% CI: 0.676–1.000, *p* = 0.011; sensitivity = 57.1%; specificity = 91.7%; Figure 1D). Regarding the serum PD-L1 levels, the optimal cut-off value in cats with HER2-positive mammary carcinoma was 2545.0 pg/mL (AUC = 0.778 ± 0.117, 95% CI: 0.548–1.000, *p* = 0.030; sensitivity = 66.7%; specificity = 92.3%; Figure 1E) and 2519.0 pg/mL in cats with TN normal-like tumor subtype (AUC = 0.857 ± 0.123, 95% CI: 0.617–1.000, *p* = 0.010; sensitivity = 85.7%; specificity = 92.3%; Figure 1F). No significant differences were found in serum PD-1 and PD-L1 levels between cats with mammary carcinoma of other molecular subtypes and the control group.

The data obtained also revealed a positive correlation between serum PD-1 and PD-L1 levels in cats with mammary carcinoma (*r* = 0.780, *p* < 0.0001, Figure 2A), particularly in those exhibiting HER2-positive (*r* = 0.786, *p* = 0.03, Figure 2B) or TN normal-like tumor subtypes (*r* = 0.857, *p* = 0.03, Figure 2C). 

### 2.2. Serum CTLA-4 and TNF-α Levels are Positively Correlated with Serum PD-1/PD-L1 Levels in Cats with HER2-Positive and TN Normal-Like Tumors

A strong positive correlation was found in cats with HER2-positive mammary carcinomas between the serum PD-1 levels and serum PD-L1 (*r* = 0.923), CTLA-4 (*r* = 0.975) and TNF-α (*r* = 0.968) levels (Table 3). The queens with TN normal-like carcinomas also showed a strong positive correlation between serum PD-1 levels and serum PD-L1 (*r* = 0.857), CTLA-4 (*r* = 0.927) and TNF-α (*r* = 0.893) levels (Table 3). 

### 2.3. TN normal-like Mammary Carcinomas Showed Lower PD-L1 Expression than HER2-Positive Tumors

Considering the high serum PD-L1 levels presented by cats with HER2-positive or TN normal-like mammary carcinomas, we further evaluated the percentage and staining intensity of PD-L1 expression in tumor-infiltrating lymphocytes (TILs) and in cancer cells, by immunostaining of tumor samples. Results showed a higher score of PD-L1-positive TILs in HER2-positive tumors than in TN normal-like ones (*p* = 0.037, Figure 3A and Figure 4A,B,E,F). Besides, the PD-L1 score on cancer cells was significantly higher in HER2-positive tumors than in TN normal-like mammary carcinomas (*p* = 0.010, Figure 3B and Figure 4C–F).

Regarding PD-1 expression in TILs and cancer cells, no significant differences were found between HER2-positive tumors and TN normal-like mammary carcinomas (*p* = 0.158, Figure 3C; *p* = 0.958, Figure 3D).

### 2.4. Elevated Serum PD-1/PD-L1 Levels Showed A Higher Concordance with IHC for PD-1 and PD-L1 in Cancer Cells than in TILs

Our results revealed that elevated serum PD-1 levels, PD-1-positive TILs and PD-1-positive cancer cells were found in 61.5%, 41.7% and 100% of cats with HER2-positive mammary carcinoma and in 57.1%, 33.3% and 100% of cats with triple-negative normal-like mammary tumors. In parallel, elevated serum PD-L1 levels, PD-L1-positive TILs and PD-L1-positive cancer cells were detected in 76.9%, 66.7% and 100% of cats with HER2-positive tumors and in 85.7%, 20% and 100% of cats with a triple-negative normal-like tumor subtype (Table 4). Thus, these results demonstrated that serum analysis could provide differential information when compared with the expression of the two immune checkpoint molecules in TILs and cancer cells. 

### 2.5. Three Non-Redundant Mutations Were Detected in Feline PD-L1 gene

Results from DNA sequencing allowed us to identify three heterozygous mutations in the feline *PD-L1* gene (Figure 5). Two were found in exon 4 (c.16.756G > A; c.16.777G >A), one changing a neutral nonpolar amino acid (alanine-245) to a neutral polar residue (threonine) in 3.8% of the studied population (1/26). The second mutation converted a methionine to a valine (p.V252M), both neutral nonpolar amino acids, in 42.3% of the animals evaluated (11/26). A third mutation was identified in exon 5 (c.18.526A > C), replacing a neutral polar threonine by a neutral polar serine at residue number 267, in 3.8% of the DNA samples (1/26).

## 3. Discussion

In order to escape the immune system, cancer cells can develop mechanisms to downregulate immune responses. For example, by overexpression of distinct immune checkpoint molecules such as PD-L1. In humans, the interaction between PD-L1 and its receptor has been revealed as an important step in the maintenance of an immunosuppressive tumor microenvironment [30]. Thus, a better understanding of the PD-1/PD-L1 pathway may also contribute to the development of new diagnostic tools and molecular therapies targeting PD-L1 in pets. 

The results obtained showed that cats presenting mammary carcinoma subtypes associated with more aggressive features and poor prognosis [31] (HER2-positive and TN normal-like) displayed significantly increased serum PD-1 and PD-L1 levels, as reported in humans with HER2-positive metastatic breast cancer [32], TN breast cancer [23], renal cell carcinoma [25], esophageal cancer [33], gastric cancer [24], advanced pancreatic cancer [34], lung cancer [35] and metastatic melanoma [36] correlated to shorter overall survival and tumor-free survival times [23,25,32,33,35], suggesting a conserved role of the PD-1/PD-L1 axis in both species. Moreover, as reported in humans, our findings uncovered a positive correlation between serum PD-1 and PD-L1 levels, suggesting that both molecules are co-regulated [23,34]. Furthermore, cats with HER2-positive or TN normal-like tumors showed a strong positive correlation between serum PD-1, PD-L1, CTLA-4 and TNF-α levels, which are immune-inhibitory molecules that downregulate T-cell immune responses, suggesting that these animals were immunosuppressed. CTLA-4 is another immune molecule located on the surface of T cells and expressed by regulatory T cells (Tregs), which triggers an inhibitory signal to immune cells [37], also being targeted in breast cancer treatment [26]. Additionally, the higher serum levels of the pro-inflammatory cytokine TNF-α may contribute to the increased tumor cell survival and PD-L1 stabilization on breast tumor cells, playing a critical role for tumor escape from immune surveillance [28]. Accordingly, a previous study also reported elevated serum CTLA-4 levels in cats with mammary carcinoma, being associated with HER2-positive status and TNF-α levels, in accordance with our findings. 

Results from the immunostaining analysis revealed that PD-L1 expression in TILs and cancer cells was higher in HER2-positive mammary carcinoma, as reported in humans [2,3,16,20,22], contrasting with TN mammary carcinoma tumor samples. In spite of our findings, previous studies in human breast cancer showed that PD-1 and PD-L1 positive TILs and cancer cells overexpressing PD-L1 were frequently found in triple negative breast cancer subtype [16,23,38,39,40,41]. These controversial results may be due to the fact that serum PD-1 levels were higher in cats with TN normal-like carcinomas compared with animals presenting HER2-positive carcinomas, indicating that sPD-1 can bind to PD-L1 attached to the cell membrane of dendritic cells, inhibiting T cell function and proliferation, as previously described [42]. Additionally, recent studies described the presence of tumor-derived exosomes that carry PD-L1, suggesting an effective paracrine mechanism that cancer cells use to communicate and reprogram immune cells and consequently suppress T-cell activation [43,44,45]. These findings support that HER2-positive cancer cells lead to a local immunosuppressive microenvironment, strengthening the systemic immunosuppression found in cats with this tumor subtype. Moreover, our results revealed PD-L1 expression in cell membrane and cytoplasm of TILs, while in cancer cells the immunostaining was distributed to cytoplasmic and nuclear membranes. Accordingly, in human breast cancer different approaches were used to evaluate the PD-L1 expression, with some studies described only membranous staining and others, in agreement with our findings, registered both membranous and cytoplasmic staining [46,47]. In addition, Ghebeh et al. showed a predominant staining in the cytoplasm near to the nuclear membrane, probably the endoplasmic reticulum, and an expression of PD-L1 in the nucleus, as we described [48]. Additionally, our results showed that elevated serum PD-1/PD-L1 levels were detected more frequently than PD-1/PD-L1 overexpression in TILs of HER2-positive and triple-negative normal-like feline mammary carcinomas. Notably, a high concordance rate was found between elevated serum PD-1/PD-L1 levels and PD-1/PD-L1 overexpression in cancer cells of the same FMC subtypes, suggesting that the serum analysis can be a non-invasive tool for the real-time assessment of these immune checkpoints molecules in feline mammary carcinoma, as reported for breast cancer [49].

Lastly, three non-redundant heterozygous mutations were found in the feline *PD-L1* gene, with none of them located at the PD-L1 extracellular domain, which is recognized by the recent and promising monoclonal antibody approved for the treatment of human metastatic TN breast cancer [29]. This shall not compromise the future development of checkpoint-blocking therapies targeting PD-L1 for FMC treatment. According to the Catalogue of Somatic Mutations in Cancer (COSMIC), six mutations were reported with very low frequency in human breast cancer patients (148/47,194; 0.3%), all different from the ones that we detected.

## 4. Material and Methods

### 4.1. Animal Population

Tumor and serum samples were collected from 53 queens with mammary carcinoma that underwent mastectomy and 15 healthy queens presenting for elective ovariohysterectomy at the Small Animal Hospital of the Veterinary Medicine Faculty, University of Lisbon. All mammary lesions were embedded in paraffin after fixation in 10% buffered formalin (pH 7.2) during 24–48 h. Serum was separated from clotted blood by centrifugation (1500× g, 10 min, 4 °C) and stored at −80 °C until further use. All samples that showed hemolysis were discarded, as recommended for humans [50].

For each animal enrolled in the study, the following clinicopathological characteristics were recorded: age, breed, reproductive status, contraceptive administration, treatment performed (none, surgery or surgery plus chemotherapy), number, location and size of tumor lesions, histopathological classification, malignancy grade, presence of tumor necrosis, lymphatic invasion, lymphocytic infiltration, cutaneous ulceration, regional lymph node involvement, stage of the disease (TNM system), disease free-survival (DFS) and overall survival (OS) [31]. Table 5 summarizes the collected clinical data. Briefly, the mean age at diagnosis was 11.8 years (range 6.5–18 years), while the mean size of the primary lesions was 2.6 cm (range 0.3–7 cm). The DFS was 9.2 ± 7.8 months (*n* = 49; 95% CI: 7.3–11.7 months) and the OS was 15.0 ± 9.8 months (*n* = 51; 95% CI: 12.2–17.7 months).

Regarding the molecular-based subtyping of FMC [31,51], cats enrolled in this study were stratified in five groups: luminal A (*n* = 8), luminal B (*n* = 18), HER2-positive (*n* = 13), TN normal-like (*n* = 7) and TN basal-like (*n* = 7).

### 4.2. Quantification of Serum PD-1, PD-L1, CTLA-4 and TNF-α Levels

Serum PD-1, PD-L1, CTLA-4 and TNF-α levels were quantified by using the commercially available immunoassay kits from R&D systems (Minneapolis, MN, USA) and following the manufacturer’s recommendations. PD-1 levels were measured with the PD-1 DuoSet ELISA kit (DY1086,); PD-L1 levels with the PD-L1/B7-H1 DuoSet ELISA kit (DY156); CTLA-4 levels with the CTLA-4 DuoSet ELISA kit (DY476); and the TNF-α levels with the TNF-α DuoSet ELISA kit (DY2586), all based on a solid-phase sandwich enzyme-linked immunosorbent assay (ELISA) technique. The concentration levels of the above molecules were calculated using appropriate standard curves, by making serial dilutions from a stock solution of recombinant PD-1, PD-L1, CTLA-4 and TNF-α (9, 18, 36, 78, 156, 312.5, 625, 1250, 2500, 5000 and 10,000 pg/mL). Serum PD-1 and PD-L1 levels (pg/mL) were calculated using a quadratic regression (y = ax^2^ + bx + c, *r*^2^ = 0.99 for PD-1 and *r*^2^ = 0.96 for PD-L1) and serum CTLA-4 and TNF-α concentrations were determined using a curve-fitting equation (y = mx + c, *r*^2^ > 0.99) [10].

Briefly, quantification of the abovementioned molecules was performed in 96-well microplates incubated overnight with the corresponding capture antibodies (100 μL/well), at room temperature RT). On the next day, plates were washed three times with 400 μL/well of PBS-Tween 0.05% and incubated with 300 μL/well of PBS/BSA blocking agent (1%, *w/v*), at RT for 60 min. Then, plates were washed three times with 400 μL/well PBS-Tween 0.05% and incubated with 100 μL/well of each dilution, in duplicate. For determination of serum PD-1, PD-L1, CTLA-4 and TNF-α levels, serum samples were diluted 20 times in PBS/BSA 1% and 100 μL of each sample was added to the wells. The plates were thereafter sealed and incubated for 2 hours at RT. After another washing step (3 × 400 μL/well PBS-Tween 0.05%), 100 μL/well of detection antibodies was added and incubated for 2 hours at RT. Later and after three washes with 400 μL/well of PBS-Tween 0.05%, 100 μL/well of streptavidin-HRP was added and incubated for 20 min at RT, avoiding placing the microplate in direct light. Afterwards a further washing step (3 × 400 μL/well PBS-Tween 0.05%), 100 μL/well of substrate solution (1:1 mixture of H2O2 and tetramethylbenzidine) was added and incubated for 20 min at RT in the dark. Finally, the reaction was stopped by adding 50 μL/well of 2 N H2SO4 and gently shaking the plate. The absorbance was measured by a spectrophotometer (Fluostar Optima Microplate Reader, BMG, Ortenberg, Germany), following the manufacturer’s recommendations.

### 4.3. Immunohistochemical Staining and Analysis

To analyze PD-1 and PD-L1 expression, two sections of formalin-fixed paraffin-embedded tissues of FMC with 3 µm thickness (Microtome Leica RM2135, Newcastle, UK) were prepared, mounted on adhesive glass slides (SuperFrost Plus, Thermo Fisher Scientific, MA, USA) and placed at 64 °C for 60 min and then at 37 °C overnight. Then deparaffinization, rehydration and epitope retrieval were performed using a PT-Link module (Dako, Agilent, Santa Clara, CA, USA), by immersing glass slides in Antigen Target Retrieval Solution pH 9 from Dako during 40 min at 95 °C. Thereafter, slides were cooled for 30 min at RT and rinsed twice for 5 min in distilled water. The endogenous peroxidase activity was blocked by an incubation period of 20 min with Peroxidase Block Novocastra Solution (Novacastra, Leica Biosystems, Newcastle, UK), and after two washing steps with PBS the nonspecific binding of immunoglobulins was prevented by incubating the tissue slices with the Protein Block Novocastra Solution (Leica Biosystems) for 60 min. Then, before two PBS washes (2 × 5 min), tissue slides were incubated with an anti-PD-1 monoclonal antibody (1:25 dilution, clone J116, Abcam, Cambridge, UK) or with an anti-PD-L1 monoclonal antibody (1:100 dilution, clone 28-8, Abcam), at 4 °C overnight. After two washes with PBS for 5 min, each tissue section was incubated at RT with the Novolink Polymer (Leica Biosystems) for 30 min. Subsequently and after an additional wash in PBS, the staining was performed using the DAB Chromogen Solution (Leica Biosystems) diluted in Novolink DAB Substrate Buffer (Leica Biosystems, 1:20, 5 min). Finally, tissue sections were counterstained with hematoxylin (Leica Biosystems) for 1 min, dehydrated in a graded ethanol series and mounted.

The analyses of PD-1 and PD-L1 immunostaining was scored following the recommendations of the International TILs Working Group 2014, established for breast cancer [52]. Briefly, the area occupied by positive lymphocytes in the stromal compartment was evaluated in whole tumor sections, excluding the TILs outside of the tumor border or in tumor zones with artifacts or necrosis, with 200–400× magnification. In addition, to identify the TILs the pathologists observed their characteristic morphology, as a scarce cytoplasm, a round and small nucleus and the presence of a dense chromatin. The scores of percentages of positive cells were recorded as: 0 (<1%), 1 (1–5%), 2 (6–30%) or 3 (>30%) [10]. The intensity scores were as follows: 0 (negative), 1+ (weak), 2+ (moderate), 3+ (strong) and 4+ (very strong). The percentage of positive cells and intensity scores were then multiplied to obtain a final IHC score. To explore the concordance between the serum PD-1/PD-L1 levels and the PD-1/PD-L1 overexpression in TILs and cancer cells, the cut-off values used were ≥5% and ≥1% for TILs overexpressing PD-1 [41,53] and PD-L1 [18], and ≥1% for cancer cells overexpressing PD-1 or PD-L1 [54,55]. Human placenta and feline lymph node tissues were used as positive controls, whereas sections of healthy mammary tissues were used as negative controls and feline lymph nodes as internal controls. All slides were independently subjected to blind scoring by two independent pathologists. 

### 4.4. DNA Extraction, Amplification and Sequence Analysis of Feline PD-L1 Gene

Genomic DNA extraction was performed in 26 primary mammary tumors using a QIAamp FFPE kit (Qiagen, Dusseldorf, Germany) following the manufacturer’s recommendations. Initially, tumor samples were homogenized with Tissue Lyser II (Qiagen) and digested with protease K in a concentration of 20 mg/mL (Qiagen). After several washing steps, the genomic DNA was eluted from the extraction columns and its quantity and quality were measured by NanoDrop ND-100 Spectrophotometer (Thermo Fischer Scientific). Then, the exons of interest in *PD-L1* gene were amplified by using specific primers (Table 6) with a PCR thermal cycler (VWR Thermocycler, Leicestershire, England). PCR procedures were performed with a standard reaction mixture (4 μL/sample of Phusion GC Buffer (Thermo Fischer Scientific), 0.4 μL/sample of dNTPs (Grisp, Porto, Portugal), 0.1 μL/sample of each forward and reverse primers and 0.2 μL/sample of DNA Polymerase (Thermo Fischer Scientific)). In all samples, PCR-grade water and DNA were added to maintain a concentration of 4 ng/mL. For amplification of exons 3 and 6, PCR reactions were performed as follows: denaturation at 98 °C for 30 s, followed by 35 cycles at 98 °C for 10 s, 56 °C for 30 s, 72 °C for 10 s, plus one extension step of 72 °C for 10 min. For exons 2, 4 and 5, the melting temperature was 60 °C. After confirmation of the expected size for each amplified fragment in an agarose gel (2.5%, Sigma-Aldrich, Darmstadt, Germany), DNA fragments were purified and sequenced by Sanger technique (StabVida, Almada, Portugal). 

The identification of PD-L1 exons was performed using BLAST software (Basic Local Alignment Search Tool, NCBI) by comparing the sequence of feline *PD-L1* gene (NC_018735.3) with its transcript (Ensembl, ENSFCAT00000078517.1). Sequenced fragments were aligned using the ClustalW tool (Bioedit software), while protein mutations and SNP loci were identified by using the Expert Protein Analysis System’s (ExPASY) translate tool (XM_006939039).

### 4.5. Statistical Analysis

The Kruskal–Wallis test and Dunn’s multiple comparisons post-test were applied to compare the serum PD-1 and PD-L1 levels between cats with different mammary carcinoma subtypes and healthy controls. Animals that had serum PD-1 or PD-L1 levels that fell more than three standard deviations from the mean were considered outliers and removed from further analysis. The optimal cut-off values were determined using the Youden index and the area under the receiver operating characteristic (ROC) curve values were calculated. Significant differences on the PD-1 and PD-L1 staining of tumor infiltrating lymphocytes and on the staining intensity of cancer cells were evaluated by using the Mann–Whitney test. Spearman’s coefficient was used to assess the correlations between serum PD-1, PD-L1, CTLA-4 and TNF-α levels.

Statistical analysis was conducted in SPSS software version 25.0 (IBM, Armonk, NY, USA) and the two-sided *p*-value < 0.05 was considered statistically significant. Results were presented as median values. GraphPad Prism version 8.1.2 (GraphPad Software, CA, USA) and Microsoft Excel for MacOS (version 16.30, Microsoft Corporation, Redmond, WA, USA) were used to plot the graphs.

## 5. Conclusions

In conclusion, our data demonstrated that serum PD-1 and PD-L1 levels were elevated in cats with HER2-positive and TN normal-like mammary carcinomas. The strong positive correlations detected between serum PD-1, PD-L1, CTLA-4 and TNF-α levels also suggest a systemic immunosuppression in cats presenting these tumor subtypes. Additionally, the microenvironment of HER2-positive tumors was revealed to be locally immunosuppressed by overexpressing PD-L1, contrasting with the human breast cancer studies that showed a higher PD-1 and PD-L1 expression in triple negative molecular subtype. Altogether, our results indicate that cats presenting HER2-positive or TN normal-like tumors could benefit from immunostimulatory therapies (e.g., anti-PD-L1, anti-CTLA-4) and provide support to the use of spontaneous feline mammary carcinoma as a model for human breast cancer.

## Figures and Tables

**Figure 1 cancers-12-01386-f001:**
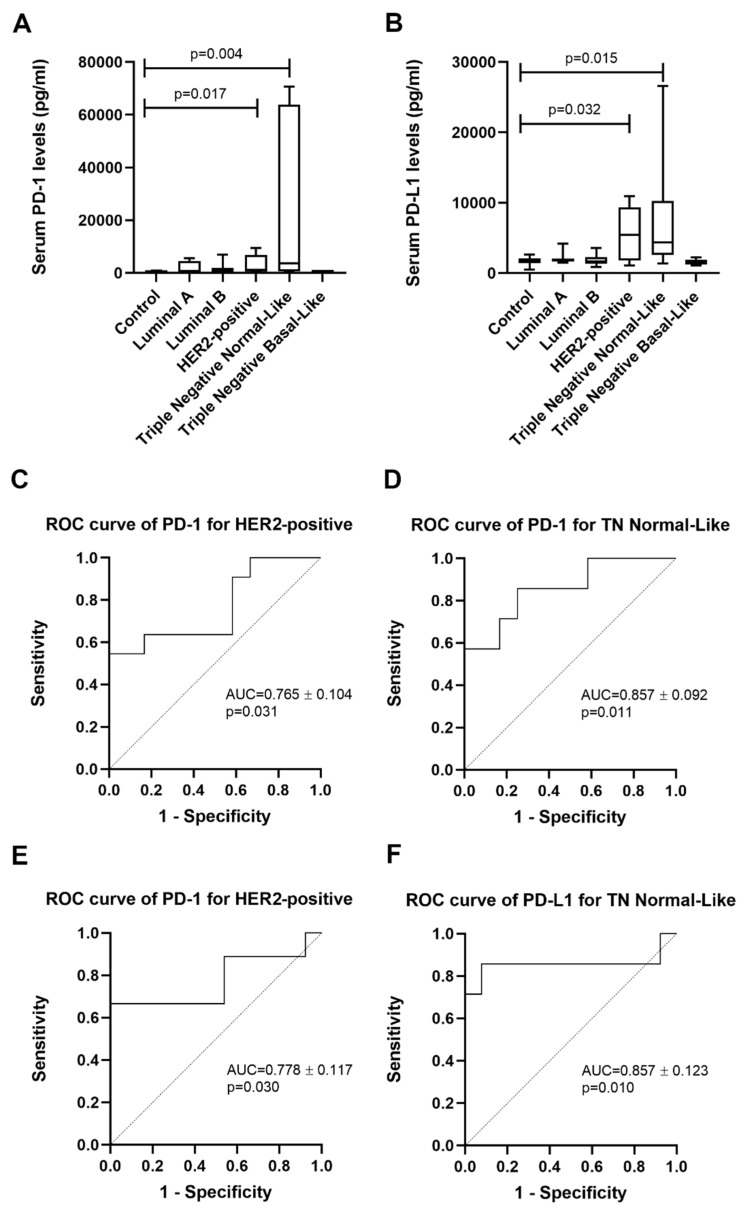
Serum programmed cell death protein-1 (PD-1) and programmed death ligand-1 (PD-L1) levels in cats with HER2-positive and triple negative (TN) normal-like mammary carcinoma. (**A**) Box plot analysis showing the range of serum PD-1 and (**B**) PD-L1 levels in control group and cats with mammary carcinoma stratified according to their molecular subtype. (**C**) Receiver Operating Characteristic (ROC) curve analysis for the identification of the optimal cut-off value of sPD-1 levels for cats with HER2-positive mammary carcinoma and (**D**) for cats with TN normal-like mammary carcinoma. (E) ROC curve analysis of PD-L1 for cats showing HER2-positive mammary carcinoma and (**F**) TN normal-like carcinoma.

**Figure 2 cancers-12-01386-f002:**
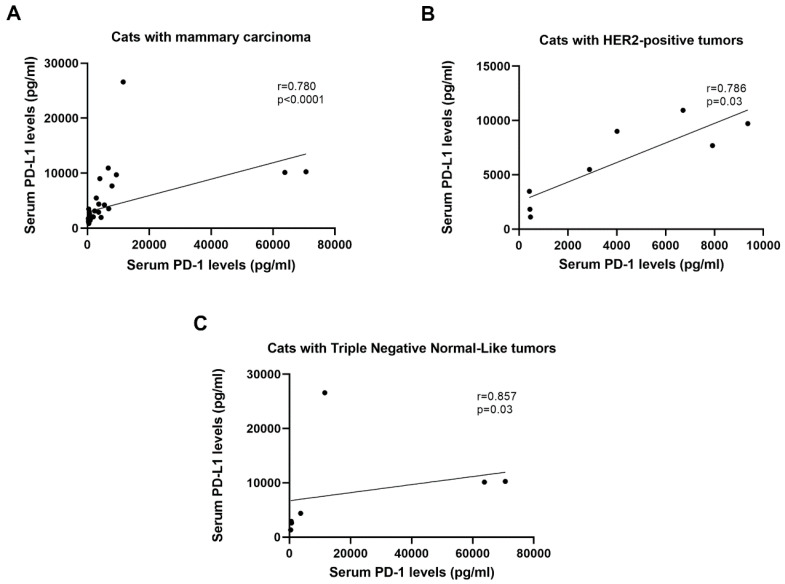
Serum programmed cell death protein-1 (PD-1) and programmed death ligand-1 (PD-L1) levels are strongly correlated in cats with HER2-positive and TN normal-like mammary carcinoma. (**A**) Correlation between sPD-1 and sPD-L1 levels in cats with mammary carcinoma (**B**) and in cats with HER2-positive and (**C**) triple negative (TN) normal-like carcinoma subtypes.

**Figure 3 cancers-12-01386-f003:**
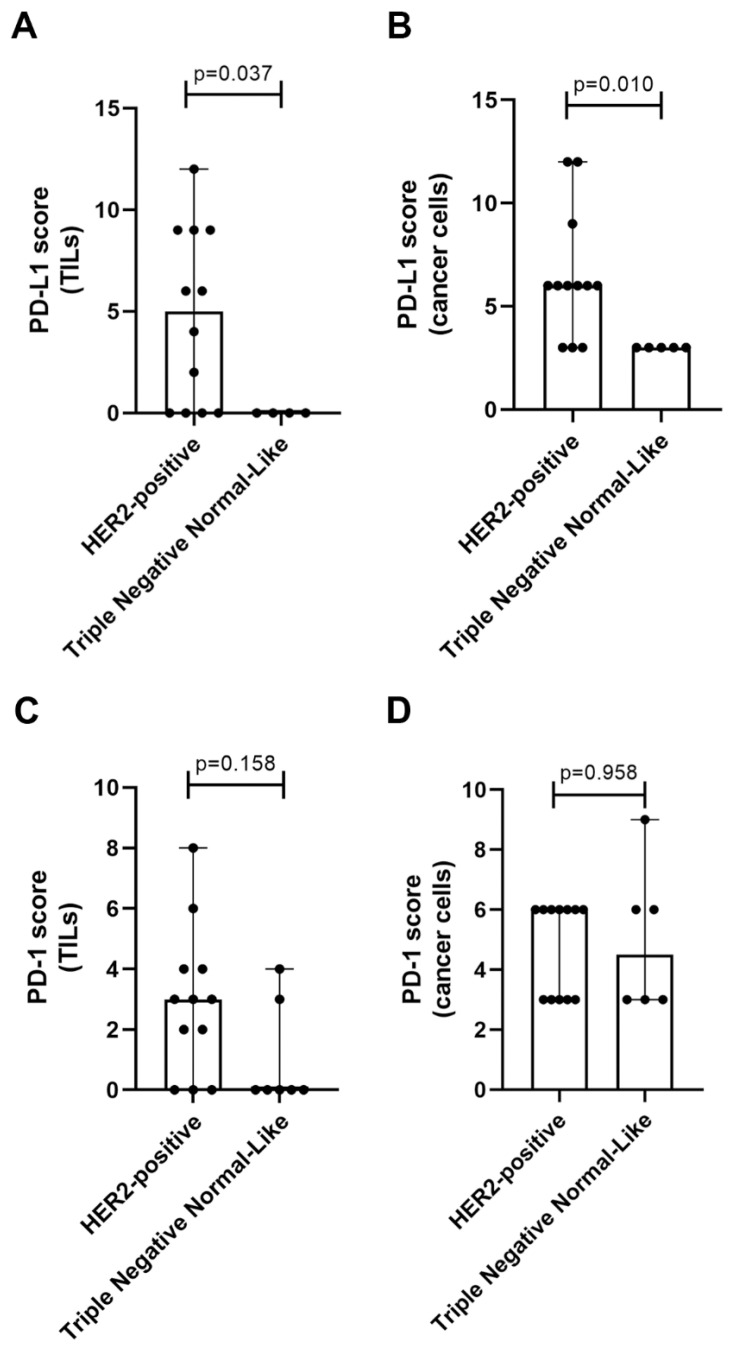
Programmed death ligand-1 (PD-L1) and programmed cell death protein-1 (PD-1) staining analysis of tumor-infiltrating lymphocytes (TILs) and cancer cells in HER2-positive and triple negative (TN) normal-like tumor samples (range values). (**A**) The score of PD-L1-positive TILs (**B**) and cancer cells in HER2-positive mammary carcinomas in comparison to TN normal-like tumors. (**C**) The score of PD-1-positive TILs between HER2-positive tumors and TN normal-like mammary carcinomas and (**D**) cancer cells.

**Figure 4 cancers-12-01386-f004:**
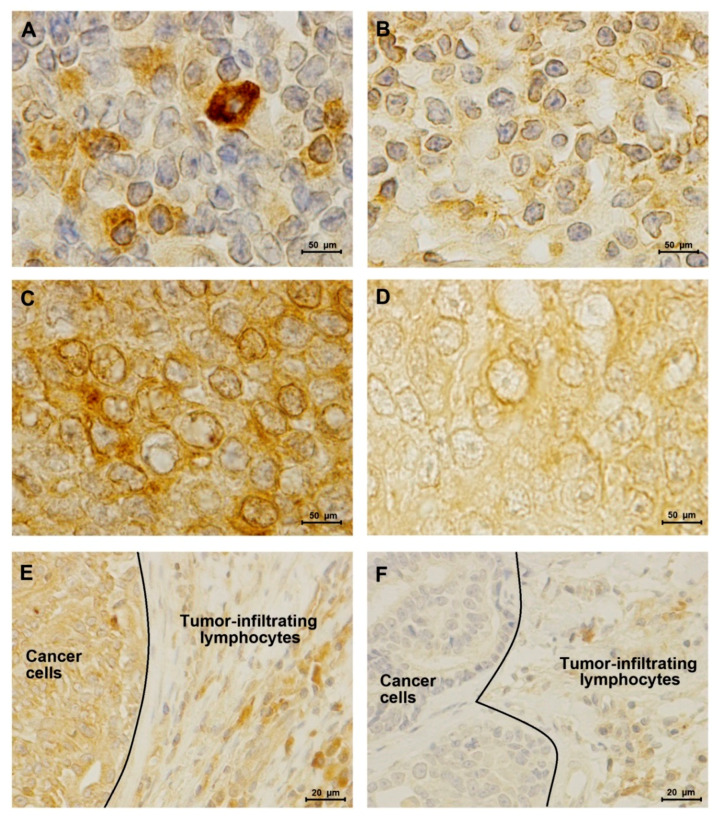
Programmed death ligand-1 (PD-L1) expression in tumor-infiltrating lymphocytes (TILs) and cancer cells of HER2-positive mammary carcinomas and triple negative (TN) normal-like tumors. (**A**) Representative PD-L1 immunostaining of tumor infiltrating lymphocytes in a HER2-positive (**B**) and in a TN normal-like mammary carcinoma (100× objective). (**C**) PD-L1 score intensity 4+ in HER2-positive cancer cells (**D**) and score 1+ in TN normal-like cancer cells (100× objective). Furthermore, in TILs, PD-L1 is localized to the cell membrane and cytoplasm, while in cancer cells it is mainly distributed to cytoplasmic and nuclear membranes. (**E**) Immunohistochemical staining of PD-L1 in a HER2-positive solid carcinoma (**F**) and in a TN normal-like tubulopapillary carcinoma (40× objective).

**Figure 5 cancers-12-01386-f005:**
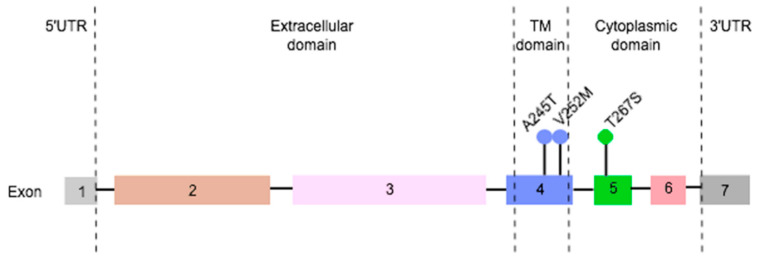
Three heterozygous and non-redundant somatic mutations were found in feline *programmed death ligand-1 (PD-L1)* gene.

**Table 1 cancers-12-01386-t001:** Serum programmed cell death protein-1 (PD-1) levels in healthy cats and queens grouped according to their mammary carcinoma molecular subtype.

Group	*n*	sPD-1 (pg/mL) Median ± SD	sPD-1 (pg/mL) Mean ± SEM
Control	12	534.0 ± 253.2	459.0 ± 73.1
Luminal A	7	708.8 ± 2165.2	1826.5 ± 818.8
Luminal B	15	590.3 ± 1776.8	1386.3 ± 458.8
HER2-positive	11	1148.9 ± 3386.0	3129.6 ± 1020.9
TN Normal-Like	7	3655.1 ± 31,478.1	21,637.3 ± 11,897.6
TN Basal-Like	6	513.3 ± 145.0	519.3 ± 59.2

**Table 2 cancers-12-01386-t002:** Serum programmed death ligand-1 (PD-L1) levels detected in healthy cats and queens stratified by their mammary carcinoma molecular subtype.

Group	*n*	sPD-L1 (pg/mL) Median ± SD	sPD-L1 (pg/mL) Mean ± SEM
Control	13	1835.5 ± 545.6	1762.5 ± 151.3
Luminal A	7	1789.6 ± 941.6	2121.5 ± 355.9
Luminal B	15	1628.0 ± 773.4	1877.6 ± 199.7
HER2-positive	9	5490.4 ± 3789.6	5672.6 ± 1263.2
TN Normal-Like	7	4377.5 ± 8819.4	8313.0 ± 3333.4
TN Basal-Like	6	1436.3 ± 425.6	1534.2 ± 173.8

**Table 3 cancers-12-01386-t003:** Spearman’s correlation between serum programmed cell death protein-1 (PD-1), programmed death ligand-1 (PD-L1), cytotoxic T-lymphocyte-associated antigen 4 (CTLA-4) and tumor necrosis factor-alpha (TNF-α) levels in cats with HER2-positive and triple negative (TN) normal-like mammary carcinomas.

**HER2-Positive**	**PD-1**	**PD-L1**	**CTLA-4**	**TNF-α**
**PD-1**	−	0.923**	0.975 **	0.968 **
**PD-L1**	0.923**	−	0.947 **	0.922 **
**CTLA-4**	0.975**	0.947**	−	0.947 **
**TNF-α**	0.968**	0.922**	0.947**	−
**TN Normal-Like**	**PD-1**	**PD-L1**	**CTLA-4**	**TNF-α**
**PD-1**	−	0.857*	0.927**	0.893**
**PD-L1**	0.857*	−	0.815*	0.857*
**CTLA-4**	0.927**	0.815*	−	0.927**
**TNF-α**	0.893**	0.857*	0.927**	−

* indicates *p* < 0.05, ** indicates *p* < 0.01.

**Table 4 cancers-12-01386-t004:** Concordance between the animals with elevated serum PD-1/PD-L1 levels (sPD-1, sPD-L1), PD-1/PD-L1–positive TILs infiltration and PD-1/PD-L1-positive cancer cells.

**Tumor subtype**	**sPD-1**	**TILs**	**Concordance**	**sPD-1**	**Cancer cells**	**Concordance**
HER2-positive	61.5%	41.7%	58.3%	61.5%	100%	66.7%
Triple Negative Normal-Like	57.1%	33.3%	0%	57.1%	100%	66.7%
**Tumor subtype**	**sPD-L1**	**TILs**	**Concordance**	**sPD-L1**	**Cancer cells**	**Concordance**
HER2-positive	76.9%	66.7%	41.7%	76.9%	100%	75%
Triple Negative Normal-Like	85.7%	20%	20%	85.7%	100%	100%

**Table 5 cancers-12-01386-t005:** Clinicopathological features of the 53 female cats with mammary carcinoma enrolled in this study.

Clinicopathological Feature	Number of Animals (%)	Clinicopathological Feature	Number of Animals (%)
Age		Tumor size	
<8 years old	3 (5.7%)	<2 cm	19 (35.8%)
≥ 8 years old	50 (94.3%)	≥ 2 cm	34 (64.2%)
Breed		HP classification	
Not determined	39 (73.6%)	Tubulopapillary carcinoma	21 (39.6%)
Siamese	5 (9.4%)	Solid carcinoma	6 (11.3%)
Persian	6 (11.3%)	Cribriform carcinoma	5 (9.4%)
Norwegian Forest Cat	2 (3.8%)	Papillary-cystic carcinoma	1 (1.9%)
Blue Russian	1 (1.9%)	Tubular carcinoma	12 (22.6%)
Spayed		Mucinous carcinoma	8 (15.2%)
No	28 (52.8%)	Tumor malignancy grade	
Yes	24 (45.3%)	I	3 (5.7%)
Unknown	1 (1.9%)	II	7 (13.2%)
Contraceptive administration		III	43 (81.1%)
No	19 (35.9%)	Tumor necrosis	
Yes	28 (52.8%)	No	13 (24.5%)
Unknown	6 (11.3%)	Yes	40 (75.5%)
Treatment		Tumor lymphatic invasion	
None	1 (1.9%)	No	48 (90.5%)
Mastectomy	48 (90.6%)	Yes	5 (9.5%)
Mastectomy + Chemo	4 (7.5%)	Lymphocytic infiltration	
Multiple tumors		No	17 (32.0%)
No	19 (35.8%)	Yes	34 (64.2%)
Yes	34 (64.2%)	Unknown	2 (3.8%)
Lymph node status		Tumor ulceration	
Negative	32 (60.4%)	No	46 (86.8%)
Positive	17 (32.1%)	Yes	7 (13.2%)
Unknown	4 (7.5%)	Ki-67 index	
TNM classification		Low (<14%)	36 (67.9%)
I	13 (24.5%)	High (>14%)	17 (32.1%)
II	6 (11.3%)	Progesterone status	
III	29 (54.7%)	Negative	28 (52.8%)
IV	5 (9.5%)	Positive	25 (47.2%)
Tumor localization		Estrogen status	
M1	8 (15.1%)	Negative	37 (69.8%)
M2	10 (18.9%)	Positive	16 (30.2%)
M3	20 (37.7%)	HER2 status	
M4	15 (28.3%)	Negative	41 (77.4%)
		Positive	12 (22.6%)

TNM—Tumor, Node, Metastasis; HP—Histopathological.

**Table 6 cancers-12-01386-t006:** Sequence of the primers used for PCR amplification of exons of feline *programmed death ligand-1* (*PD-L1*) gene.

Exons	Forward 5’–3’	Reverse 5’–3’
2	TTTGGGGACAGCAGCTTGTT	TGAACAGACTGACACCGTGG
3	TCTGAGAACCAGCCAGAATTGA	ACTGGAACATAGGGCGTGTT
4	GTCGAAGGCATCTCGCTGT	AGAGCCACTGTGACAACAACA
5	AATTGACCTCAGGGGTTGGAA	GAGGTAAGGAGGAGCCCGTT
6	TACTGCAGAGGTAACTGGACA	GGCCTCTCACATCCGACATC

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
