# Peer review of "Serum PD-1/PD-L1 Levels, Tumor Expression and PD-L1 Somatic Mutations in HER2-Positive and Triple Negative Normal-Like Feline Mammary Carcinoma Subtypes"

_cancers, 2020, doi:10.3390/cancers12061386_

Round 1

Reviewer 1 Report

Dear Authors, congratulations for your work.

Please, kindly find some comments below:

*Abstract: some acronyms are not defined (i.e., TILs, TNF-α): please check the Journal rules for this

*Introduction:

-please, provide a ref for the sentence in page 1, lines 41, 42 and 43. Remarkably PD-L1 expression is rarely expressed by tumor cells in BC (some refs: 1) Front Immunol. 2017 Oct 30;8:1412. doi: 10.3389/fimmu.2017.01412. eCollection 2017; 2) Oncoimmunology. 2016 Dec 14;6(1):e1257452. doi: 10.1080/2162402X.2016.1257452. eCollection 2017)

-line 45, page 2: please specify that PD-L1 should be expressed by Immune Cells for considering a TNBC pt candidate to atezolizumab (anti-PD-L1)

-line 64, page 2: please add (TILs) to tumor-infiltrating lymphocytes

Overall, I would spend more words on the definition of the molecular subtypes of interest to this study (TN-normal like, TN-basal like and HER2-positive).

*Results

-page 4: please check the titles of the graphs where you write HER instead of HER2.

-page 6, lines 125-129: it is not clear whether you used % for evaluating the expression of PD-L1 by TILs and intensity for the evaluation of the expression of PD-L1 by tumor cells. Please, specify. Further why didn't you consider TN-basal like in the graphs and in the analyses?

*Discussion

-line 164, page 8: I would define PD-L1 an immune checkpoint molecule

-I found it strange the statement in lines 179-182 

-line 183, page 8: please define Tregs

-line 189, page 8: please, consider adding more refs to 22 and 23 (i.e., Front Immunol. 2017 Oct 30;8:1412. doi: 10.3389/fimmu.2017.01412. eCollection 2017.)

I would spend more words on referring to the different findings observed in human breast cancer, where PD-1 and PD-L1 were found more frequently expressed in the TN subtype

*Conclusion

-why not referring to findings in humans, where PD-L1 and PD-1 are more frequently expressed in TNBC?

Author Response

Reviewer #1 (Comments to the Author):

Dear Authors, congratulations for your work. Dear reviewer, thank you so much for your positive opinion about this manuscript, we also appreciate your very valuable review.

Please, kindly find some comments below:

Abstract: some acronyms are not defined (i.e., TILs, TNF-α): please check the Journal rules for this. As suggested, the acronyms PD-1, PD-L1, CTLA-4, TNF-α and TILs were defined in the abstract (lines 19-22), as well as in the main text (lines 58, 61, 63, 72 and 79), in the figure captions and in the tables (lines 99, 101, 119, 137, 149-151, 165, 166, 174, 175, 194 and 366).

*Introduction

-please, provide a ref for the sentence in page 1, lines 41, 42 and 43. Remarkably PD-L1 expression is rarely expressed by tumor cells in BC (some refs: 1) Front Immunol. 2017 Oct 30;8:1412. doi: 10.3389/fimmu.2017.01412. eCollection 2017; 2) Oncoimmunology. 2016 Dec 14;6(1):e1257452. doi: 10.1080/2162402X.2016.1257452. eCollection 2017). As recommended, three references were provided supporting the sentence in page 2 (line 59).

-line 45, page 2: please specify that PD-L1 should be expressed by Immune Cells for considering a TNBC pt candidate to atezolizumab (anti-PD-L1). This information was added in lines 59-62. Thank you for this insightful comment.

-line 64, page 2: please add (TILs) to tumor-infiltrating lymphocytes. Added in line 90.

Overall, I would spend more words on the definition of the molecular subtypes of interest to this study (TN-normal like, TN-basal like and HER2-positive). Further information was added on the molecular breast cancer subtypes (lines 39-44) and on the association between the PD-1 and PD-L1 expression and tumor malignancy (lines 66 and 67).

*Results

-page 4: please check the titles of the graphs where you write HER instead of HER2. Graphs were corrected (Figures 1C and 1E).

-page 6, lines 125-129: it is not clear whether you used % for evaluating the expression of PD-L1 by TILs and intensity for the evaluation of the expression of PD-L1 by tumor cells. Please, specify. As suggested, the two scoring methods were clarified in line 155. Further why didn't you consider TN-basal like in the graphs and in the analyses? We took this decision based on the results obtained from the serum samples, since no significant differences were found in serum PD-1 and PD-L1 levels between healthy cats and cats with TN basal-like mammary carcinomas.

*Discussion

-line 164, page 8: I would define PD-L1 an immune checkpoint molecule. Corrected in line 200.

-I found it strange the statement in lines 179-182. Knowing that PD-1, PD-L1 and CTLA-4 are immune-checkpoint molecules that downregulate T-cell immune responses and that TNF-α is an autocrine growth factor constitutively express by cancer cells, we postulated that cats with elevated serum PD-1, PD-L1, CTLA-1 and TNF-α are immunosuppressed. Accordingly, the statement was better clarified (lines 217 and 218).

-line 183, page 8: please define Tregs. Tregs were defined in line 220.

-line 189, page 8: please, consider adding more refs to 22 and 23 (i.e., Front Immunol. 2017 Oct 30;8:1412. doi: 10.3389/fimmu.2017.01412. eCollection 2017.) As recommended, three additional references were added, including the one suggested by the reviewer (line 229).

I would spend more words on referring to the different findings observed in human breast cancer, where PD-1 and PD-L1 were found more frequently expressed in the TN subtype. More information on this topic was added in lines 229-232.

*Conclusion

-why not referring to findings in humans, where PD-L1 and PD-1 are more frequently expressed in TNBC? As requested, additional information was added in lines 394 and 395.

Reviewer 2 Report

Authors examined the serum levels of CTLA-4, TNF-alpha, PD1 and PDL1 in cats with different subtypes of breast cancer and compared to healthy controls, as well as PDL1 gene sequencing. The study showed that serum PD 1 and PDL1 levels were significantly higher in cats with HER2 pos and TN normal-like mammary carcinoma, and there is a positive correlation between serum PD1, PDL1, CTLA-4 and TNF alpha.  PDL1 expression in cancer cells was significantly higher in HER2 pos samples than TN normal like tumors. There were mutations identified in PDL1 sequencing.  Authors concluded that PD-1 and PDL1 levels can be used as diagnostic biomarkers of HER2 pos and TN normal like feline mammary carcinoma.  Concerns about the study is that it is unclear whether tumor stage influenced the results.  In addition, the study is largely descriptive and lack of functional studies. It is unclear whether the mutations would be functionally relevant.

Author Response

Authors examined the serum levels of CTLA-4, TNF-alpha, PD1 and PDL1 in cats with different subtypes of breast cancer and compared to healthy controls, as well as PDL1 gene sequencing. The study showed that serum PD 1 and PDL1 levels were significantly higher in cats with HER2 pos and TN normal-like mammary carcinoma, and there is a positive correlation between serum PD1, PDL1, CTLA-4 and TNF alpha.  PDL1 expression in cancer cells was significantly higher in HER2 pos samples than TN normal like tumors. There were mutations identified in PDL1 sequencing.  Authors concluded that PD-1 and PDL1 levels can be used as diagnostic biomarkers of HER2 pos and TN normal like feline mammary carcinoma.  Concerns about the study is that it is unclear whether tumor stage influenced the results.  Dear reviewer, thank you for this relevant remark. The statistical analysis revealed that the tumor stage did not show a significant correlation with the tumor molecular subtype (p=0.139, data not shown), meaning that this clinicopathological feature does not influence the results and the conclusions.

In addition, the study is largely descriptive and lack of functional studies. Dear reviewer, although we agree with your comment, we would like to remind you that this work results from a follow-up study of 5 years that enrolled a large number of queens with mammary carcinoma, a spontaneous tumor that shows similar biological behavior to human breast cancer when compared with the chemically induced and/or genetically engineered mouse mammary cancer models. Furthermore, please keep in mind that this is the first study that evaluates the serum PD-1/PD-L1 levels and the PD-1/PD-L1 expression in tumor-infiltrating lymphocytes in pets, and, consequently, in cats which were stratified by the mammary carcinoma molecular. Indeed, this study opens new avenues of investigation that could ultimately lead to the development of diagnostic tools and PD-L1-directed molecular treatments that may improve overall survival and/or the disease-free survival in cats with mammary carcinoma. As a direct consequence of this work, we are currently working with a research group that has expertise in antibody development and production, to engineering a feline anti-PD-L1 antibody. Finally, the results herein reported, also reinforce the hypothesis that the spontaneous feline mammary carcinoma is a suitable model for comparative oncology studies, anticipating a high number of citations.

It is unclear whether the mutations would be functionally relevant. Dear reviewer, thank you for raising this very interesting point. Unfortunately, we are not able to demonstrate if the three point mutations identified are functionally irrelevant (in vitro studies will be necessary). However, no mutations were found in the extracellular domain of the PD-L1, suggesting that the binding capacity of the PD-L1 to PD-1 receptor is unaffected, in the studied population, as well as, the binding affinity of a putative therapeutic anti-PD-L1 antibody. Indeed, in two of the three mutations identified, the original amino acid residue was replaced by another with similar biophysical proprieties (V252M, both AAs are nonpolar and the mutation is in the transmembrane domain and is present in 42,3% of the animals analyzed; T267S, both AAs are polar and the mutation is in the cytoplasmic domain of the PD-L1, in 3,8% of the cats analyzed). Notably, just in the point mutation A245T, the original amino acid residue (alanine, non-polar) is replaced by AAs with different charge proprieties (threonine, polar). Finally, none of these mutations were identified/characterized in humans or reported in the COSMIC database or ICGC/TCGA Pan-Cancer project. 

Reviewer 3 Report

In the current study, molecules associated with immune evasion were investigated in the blood and tumor tissue of cats with mammary carcinoma. It is a well conducted study providing interesting findings. However the impact of the findings is not clear. The authors should provide some additional information on the therapeutic strategies commonly being applied and on the potential impact of immunotherapy in the management of breast cancer in cats. Also, it should be more clearly presented if cats could serve as a representative animal model to investigate immune suppression in breast cancer. Previous evidence regarding the expression of the indiviual markers in plasma and/or tissue of cats should be also provided and accordingly discussed.

Major comments

  1. The discrepancy regarding the PD-1 and PD-L1 assessment among TILs and cancer cells makes the results incomplete. TILs were characterized by using the percentage of expression only, whereas, cancer cells were characterized using the intensity score. I would suggest to the authors to use the same scoring approach among the two individual cell populations, which could be a combination of percentage and intensity. Nevertheless, the already observed differences in the percentage of TILs and in the intensity of cancer cells could be additionally presented.
  2. The authors demonstrate that " ... in TILs, PD-L1 is localized to the cell membrane and cytoplasm, while in cancer cells is mainly distributed to cytoplasmic and nuclear membranes". These findings should be discussed in the context of the previously reported PD-L1 localization in TILs and cancer cells, which is different in humans. Please provide any available data from cats.
  3. I would suggest to rephrase the sentence (lines 176-179) "Moreover, the determined serum PD-1 and PD-L1 cut-off values showed high sensitivity and specificity, indicating that these molecules can be used as promising biomarkers for the diagnosis of HER2-positive and TN normal-like mammary carcinomas". This conclusion is not supported by the results, since the sensitivity was low, and importantly, correlations observed do not support the diagnostic role of the markers regarding the molecular subtype. This should be also corrected in the Conclusions Section.
  4. "It has also been suggested that serum PD-L1 levels can result from the release of stromal and cancer cells to the bloodstream [5], justifying the higher median of sPD-L1 levels found in cats with HER2-positive tumors than those showing TN normal-like mammary carcinomas." The authors should investigate possible correlations among sPD-1 or sPD-L1 and PD-L1 expression in the tumor tissue of HER2-positive and TN normal-like cats.

Minor comments

  1. Regarding the values of sPD-1 and sPD-L1 in Tables 1 and 2, SD (standard deviation) corresponds to the median values, whereas, SEM (standard error of mean) should be presented for the mean values instead.
  2. Lines 106-108 and Figure 2, linear regression for HER2-positive and TN normal-like disease should be individually presented.
  3. In Figure 1 and Figure 3, it should be mentioned which values are represented by the lines and the error bars (or range?) in the box plots presented.
  4. I would suggest that figure legends would include only the description of the figure rather than results and p values.
  5. In Figure 4, arrows or symbols could be used to discriminate TILs from cancer cells.
  6. Lines 188-190 "Results from the immunostaining analysis revealed that in HER2-positive mammary carcinoma, TILs and cancer cells overexpress PD-L1, as reported in humans [22,23], contrasting with TN mammary carcinoma tumor samples.". Please rephrase it is not clear. Also provide the discordant evidence supporting the increased PD-L1 expression in TN disease.
  7. In Methods, the approach used for the characterization of TILs should be also provided.

Author Response

Reviewer #3 (Comments to the Author):

In the current study, molecules associated with immune evasion were investigated in the blood and tumor tissue of cats with mammary carcinoma. It is a well conducted study providing interesting findings. However the impact of the findings is not clear. The authors should provide some additional information on the therapeutic strategies commonly being applied and on the potential impact of immunotherapy in the management of breast cancer in cats. As suggested, additional information was added on current therapeutic strategies in cats with mammary carcinoma (lines 46-48).

Also, it should be more clearly presented if cats could serve as a representative animal model to investigate immune suppression in breast cancer. As recommended, further information was added to clarify the utility of feline mammary carcinoma as a cancer model (lines 50-56, 397 and 398).

Previous evidence regarding the expression of the individual markers in plasma and/or tissue of cats should be also provided and accordingly discussed. As suggested, more information on this topic was added in lines 74, 75, 81, 82, 224-226.

Major comments

  1. The discrepancy regarding the PD-1 and PD-L1 assessment among TILs and cancer cells makes the results incomplete. TILs were characterized by using the percentage of expression only, whereas, cancer cells were characterized using the intensity score. I would suggest to the authors to use the same scoring approach among the two individual cell populations, which could be a combination of percentage and intensity. Nevertheless, the already observed differences in the percentage of TILs and in the intensity of cancer cells could be additionally presented. As requested, the expression of PD-1 and PD-L1 was evaluated in cancer cells and in TILs using the same scoring method, which combines the percentage of positive cells and the staining intensity (please see Material and Methods, lines 341-344). Although the results obtained are similar to those previously reported, a new figure 3 was prepared and the results were updated (lines 29, 157, 158, 159, 162, 168 and 170).

2. The authors demonstrate that " ... in TILs, PD-L1 is localized to the cell membrane and cytoplasm, while in cancer cells is mainly distributed to cytoplasmic and nuclear membranes". These findings should be discussed in the context of the previously reported PD-L1 localization in TILs and cancer cells, which is different in humans. Please provide any available data from cats. Dear reviewer, to the best of our knowledge this is the first study to evaluate the expression of PD-L1 in TILs and cancer cells in queens, so we couldn’t provide any data from previous studies in cats. However, as you suggested, we discussed a bit more in depth this issue in lines 243-250, contrasting the distribution pattern of PD-L1 between humans and cat. Thank you for this insightful suggestion.

3. I would suggest to rephrase the sentence (lines 176-179) "Moreover, the determined serum PD-1 and PD-L1 cut-off values showed high sensitivity and specificity, indicating that these molecules can be used as promising biomarkers for the diagnosis of HER2-positive and TN normal-like mammary carcinomas". This conclusion is not supported by the results, since the sensitivity was low, and importantly, correlations observed do not support the diagnostic role of the markers regarding the molecular subtype. This should be also corrected in the Conclusions Section. Thank you for this suggestion. This sentence was removed from the abstract and discussion sections. Additionally, the authors rephrased lines 389 and 390 in the conclusions section.

4. "It has also been suggested that serum PD-L1 levels can result from the release of stromal and cancer cells to the bloodstream [5], justifying the higher median of sPD-L1 levels found in cats with HER2-positive tumors than those showing TN normal-like mammary carcinomas." The authors should investigate possible correlations among sPD-1 or sPD-L1 and PD-L1 expression in the tumor tissue of HER2-positive and TN normal-like cats. Dear reviewer, thank you for raising this very interesting point. Results from statistical analysis revealed that serum PD-1 or PD-L1 levels did not show a significant correlation with the PD-L1 expression in the tumor tissue of HER2-positive or TN normal-like subtypes (p>0.05). Therefore, this sentence was removed from the discussion section.

HER2-positive

Score PD-L1 (cancer cells)

Score PD-L1 (TILs)

sPD-1

0.373 (p=0.232)

-0.316 (p=0.317)

sPD-L1

0.320 (p=0.310)

-0.169 (p=0.600)

TN normal-like

Score PD-L1 (cancer cells)

Score PD-L1 (TILs)

sPD-1

-0.131 (p=0.805)

-0.354 (p=0.559)

sPD-L1

-0.131 (p=0.805)

-0.707 (p=0.182)

Minor comments

  1. Regarding the values of sPD-1 and sPD-L1 in Tables 1 and 2, SD (standard deviation) corresponds to the median values, whereas, SEM (standard error of mean) should be presented for the mean values instead. Corrected.

  1. Lines 106-108 and Figure 2, linear regression for HER2-positive and TN normal-like disease should be individually presented. As suggested, distinct graphs with linear regression were added for HER2-positive mammary carcinomas (figure 2B) and for TN normal-like tumors (figure 2C). The results have been updated accordingly (lines 135, 136 and 140).

  1. In Figure 1 and Figure 3, it should be mentioned which values are represented by the lines and the error bars (or range?) in the box plots presented. The values are mentioned in the captions of figures 1 and 3.

  1. I would suggest that figure legends would include only the description of the figure rather than results and p values. Figure captions were revised as suggested.

  1. In Figure 4, arrows or symbols could be used to discriminate TILs from cancer cells. In figures 4E e 4F, a line was added to discriminate cancer cells from TILs, also identified.

  1. Lines 188-190 "Results from the immunostaining analysis revealed that in HER2-positive mammary carcinoma, TILs and cancer cells overexpress PD-L1, as reported in humans [22,23], contrasting with TN mammary carcinoma tumor samples.". Please rephrase it is not clear. Also provide the discordant evidence supporting the increased PD-L1 expression in TN disease. The sentence was rephrased to be clearer (lines 227 and 228). Additionally, as recommended, more information was added about the fact that PD-1 and PD-L1 overexpression is frequently detected in triple negative human breast cancer (lines 229-232).

  1. In Methods, the approach used for the characterization of TILs should be also provided. This information was added to the methods section (lines 339-341).

Round 2

Reviewer 2 Report

Authors have addressed my concerns.

Author Response

Reviewer #2 (Comments to the Author):

Authors have addressed my concerns. Dear reviewer, thank you for your previous comments and suggestions. All the authors appreciate your valuable review.

Reviewer 3 Report

The authors have properly modified the manuscript and sufficiently answered all my queries.

The only point that I find interesting to be added is the discrepancy of PD-1 and PD-L1 expression among the blood and the cancer tissue. In which compartment were the two markers more frequently expressed? It would be interesting to show the percentage of cats with discordant results among the two compartments. In my opinion, it is not a negative result, instead it is an interesting finding suggesting that the blood analysis may provide different information regarding the expression pattern of the two immune checkpoints. In accordance, recent evidence in patients with breast cancer supports that PD-L1 expression in both the immune cells and cancer cells, significantly differs among the peripheral blood and tumor tissue, indicating that blood analysis could serve as a non-invasive tool for the real-time assessment of immune checkpoints in breast cancer (doi: 10.3390/cancers12020376).

My suggestion is to modify the title of the manuscript in order to be representative of the findings. The study does not support that PD-1 and PD-L1 can serve as diagnostic biomarkers in cats, or that PD-L1 should be therapeutically targeted in cats. I would rather suggest "PD-1 and PD-L1 expression in HER2-positive and triple negative normal-like feline mammary carcinoma subtypes", or any other general title.

Author Response

Reviewer #3 (Comments to the Author):

The authors have properly modified the manuscript and sufficiently answered all my queries. Dear reviewer, thank you very much for your insightful comments and constructive remarks along the manuscript. They really improve the final quality of the manuscript.  

The only point that I find interesting to be added is the discrepancy of PD-1 and PD-L1 expression among the blood and the cancer tissue. In which compartment were the two markers more frequently expressed? It would be interesting to show the percentage of cats with discordant results among the two compartments. In my opinion, it is not a negative result, instead it is an interesting finding suggesting that the blood analysis may provide different information regarding the expression pattern of the two immune checkpoints. In accordance, recent evidence in patients with breast cancer supports that PD-L1 expression in both the immune cells and cancer cells, significantly differs among the peripheral blood and tumor tissue, indicating that blood analysis could serve as a non-invasive tool for the real-time assessment of immune checkpoints in breast cancer (doi: 10.3390/cancers12020376). As recommended, new results were presented in lines 168-177 and in Table 4, discussed in lines 238-244 and in M&M section (333-336). The authors would like to thank you once again for your suggestions that greatly improved this work.

My suggestion is to modify the title of the manuscript in order to be representative of the findings. The study does not support that PD-1 and PD-L1 can serve as diagnostic biomarkers in cats, or that PD-L1 should be therapeutically targeted in cats. I would rather suggest "PD-1 and PD-L1 expression in HER2-positive and triple negative normal-like feline mammary carcinoma subtypes", or any other general title. As suggested, the title was modified to “Serum PD-1/PD-L1 levels, tumor expression and PD-L1 somatic mutations in HER2-positive and triple negative normal-like feline mammary carcinoma subtypes”.